# A systematic review of the 60 year literature: Effects of outreach programs in supporting historically marginalized and first-generation, low-income students in healthcare education

Eric Jenkins[☯], Jocelyn Elizabeth Nardo[ORCID]*[☯], Shima Salehi

Graduate School of Education, Stanford University, Stanford, California, United States of America

☯ These authors contributed equally to this work.
* jnardo@stanford.edu

**Data Availability Statement:** All relevant data are within the paper and its Supporting Information files.

## Abstract

We have reviewed over 60 years of studies on healthcare education outreach programs that are aimed to support first-generation, low-income, as well as underrepresented racial and ethnic minority groups (historically marginalized students) to pursue pre-health professions. As a systematic literature review, we present the challenges studies on healthcare education outreach programs had as three main categories: 1) **Design**, 2) **Evaluation**, and 3) **Analysis**. 1) **Designs** of studies on healthcare education outreach programs often lacked theoretical foundations whereby a) the interventions did not present theories underlying a causal mechanism of inequity in health professions; and/or 1b) the defined outcome measures were not clearly aligned with the problem the intervention tried to address. 2) **Evaluations** of studies on healthcare education outreach programs were not always conducted effectively whereby: 2a) controlled groups were commonly absent for comparison with the intervention group; and/or 2b) post measures were solely used without pre-measures. 3) **Analyses** of studies on healthcare education outreach programs were not adequate whereby: 3a) the response rates and effect size were commonly low; and/or 3b) qualitative results commonly did not supplement quantitative results. Overall, our findings reveal studies on healthcare education outreach programs have common challenges that hinder the reliability of their effects supporting historically marginalized students in pursuing pre-health professions. To address such challenges with studies on healthcare education outreach programs aimed at supporting historically marginalized students, we created a decision flow chart for researchers to ask themselves: 1) how is the design guided by theoretical goals; 2) how are measurements used to evaluate success; and 3) how does the analysis lead to reliable results?

**Funding:** The author(s) received no specific funding for this work.

**Competing interests:** The authors have declared that no competing interests exist.

## Introduction

Originating from long-standing social inequity, healthcare disparity in the United States for first-generation Americans, persons from low socioeconomic backgrounds, and underrepresented racial and ethnic minority groups is growing especially in light of the COVID-19 pandemic [1]. Healthcare disparity is impacted by healthcare professionals' ability to empathize with patients, which can be limited by compassion burnout [2] and cultural competency [3]. However, healthcare professionals who are first-generation Americans, persons from low socioeconomic backgrounds, and Black, Indigenous, People of Color (BIPOC) have greatly contributed to how medical conditions such as strokes, heart attacks, and cancer impact sub-populations differently, which reflects how compassion and cultural competency often originate at the intersection of lived experience and medical training [4]. Thus, researchers have hypothesized healthcare disparity can be addressed by supporting students who are first-generation Americans, persons from low socioeconomic backgrounds, and underrepresented racial and ethnic minorities (hereafter, historically marginalized students) to become healthcare professionals [4] since historically marginalized students are often multilingual [5], have financial knowledge from navigating the healthcare system from the position of a person from a low socioeconomic background [6] and have the necessary compassion and cultural competency [7]. Unfortunately, a massive *education disparity* also exists for historically marginalized students as compared to their historically non-marginalized counterparts to pursue healthcare professions [8].

Many private philanthropic organizations, academic systems, and federal or state governments have allocated money and resources for recruitment of historically marginalized students into medical, dental, pharmaceutical, and other post-secondary programs to mitigate the education disparity [9, 10]. Examples of recruitment strategies for healthcare programs consist of soliciting and recruiting students from diverse areas of the United States to pursue healthcare professions by waiving application fees [11] or lowering qualifying exam scores [12]. The argument for recruitment-based strategies derives from examining the *pipeline* of healthcare professionals [13]. Essentially, the pipeline metaphor posits that creating a more diverse healthcare, and in general STEM, workforce necessitates plugging the metaphorical "leaks." The leaks refer to the loss of historically marginalized students interested in these professions throughout their education pathways [14]. Yet, nearly 60 years of active recruitment of historically marginalized groups into post-secondary programs to address the education disparity have still not addressed the growing underrepresentation of such groups within healthcare professions, resulting in the current healthcare disparity [15, 16].

Therefore, the pipeline metaphor for creating a more diverse STEM workforce should be critically elaborated by considering how institutional, societal, and cultural factors shape the trajectories of historically marginalized students [14, 17–20]. Moving away from the pipeline metaphor that focuses mostly on recruitment [14], research has suggested providing resources, networks, and opportunities early on through healthcare education outreach programs for the intrinsic and societal benefits of historically marginalized students [21–23]. Healthcare education outreach programs have been shown to facilitate the educational pathways of historically marginalized students to pursue STEM majors, leading to long-term careers in healthcare professions [23]. For example, Stray and colleagues found that the ongoing (1992-present) collaboration between faculty mentors at the University of Mississippi Medical Center (UMMC) and local high school teachers and students increased high school graduation rates, college application submissions, enrollment in STEM baccalaureate majors, and continued STEM enrollment for master's degrees as well as degrees in healthcare professions [22]. This was due to UMMC's support of the local community, mentorship of science teachers, and inquiry-

based classroom activities that were funded by UMMC. The study at UMMC drew from the Mississippi Science Framework that theorized program components such as student research mentorship, teacher professional development, disciplinary content knowledge enrichment, and practical experiences, which has promoted sustainability of the program for 28 years [22].

Hence, the structure of effective healthcare education outreach programs is an emerging direction for researchers to pursue in addressing the education and resulting healthcare disparity in the United States. To understand how healthcare education outreach programs are researched, we employed a systematic literature review [24] to examine studies on healthcare education outreach programs across six decades. We examined how studies on healthcare education outreach programs were designed, evaluated, and analyzed to support historically marginalized students reliably. The review suggests that although the emergent direction can be productive for supporting historically marginalized students to pursue and persist in healthcare professions, there is a notable paucity of evaluated research that can provide sufficient evidence to guide the structure of healthcare education outreach programs [24]. We hope by providing a systematic literature review, we can support researchers structuring and facilitating healthcare education outreach programs that can reliably determine efficacious outcomes for historically marginalized students to pursue future STEM degrees, leading eventually to healthcare professions and dismantling healthcare disparity in the United States. Our systematic literature review seeks to answer:

- How are studies on healthcare education outreach programs designed by drawing from theory?

- How are studies on healthcare education outreach programs evaluated using quantitative, qualitative, and mixed-methods?

- How are studies on healthcare education outreach programs analyzed to generate reliable and generalizable results?

## Overview of current review methods

The following literature review is considered a systematic literature review [25], which contains these important features to search and evaluate the literature: eligibility criteria; inclusion criterion; search strategy; literature identification; selection process; data collection; screening for inclusion; data items; quality and eligibility assessment; study risk of bias assessment; reporting bias assessment; effect measures; synthesis methods; and certainty assessment [26]. Table 1 groups alike features together and describes how the feature works in the context of our systematic literature review.

### Study search, retrieval, eligibility, and selection

All three authors initially searched the following databases: ERIC, American Educational Research Journal, Education Research Complete, MedEdPORTAL, SCOPUS, Web of Science, and Academic Search Premier within Stanford's Lane Library combined database. Within each database, we searched within the last ten years and used the following search terms: mentor* AND (STEMM or STEM or healthcare OR "healthcare" OR medic* OR "health science*") AND ("high school" OR "secondary school") AND (minority* OR black OR latin* OR "first gen*" OR "first-gen*" OR underrepresented) AND "outreach" OR "pipeline program." However, only 19 articles were generated within the last ten years. Due to the small initial turn-out of articles related to the keywords, the search was expanded to include *all* articles regardless of publication date. Our second search returned 55 relevant articles from the

**Table 1. Features of a systematic literature review and context.**

| Features | Definition of Feature | Context |
|---|---|---|
| Eligibility Criteria<br>Inclusion Criterion | The breadth of the literature that the authors choose to review. | Our systematic literature review draws from science education, health-related journals, and cultural studies in education that are concerned with healthcare education outreach programs. To be considered, the studies selected must be peer-reviewed and written in English. |
| Search Strategy<br>Literature Identification | The databases used to search the literature that the authors choose to review. | We searched within ERIC, American Educational Research Journal, Education Research Complete, MedEdPORTAL, SCOPUS, Web of Science, Academic Search Premier within Stanford's Lane Library combined database. |
| Selection Process<br>Data Collection<br>Screening for Inclusion | The keywords for consideration that the authors choose to review. | We searched used the databases using the keywords: mentor* AND (STEMM or STEM or healthcare OR "healthcare" OR medic* OR "health science*") AND ("high school" OR "secondary school") AND (minority* OR black OR latin* OR "first gen*" OR "first-gen*" OR underrepresented) AND "outreach" OR "pipeline program." |
| Data Items<br>Quality and Eligibility Assessment | The process for ensuring that the articles selected from the literature meet the inclusion criterion, literature identification, and screening for inclusion. | All three authors were involved in reviewing the abstracts to ensure selected studies contained either qualitative, quantitative, or mixed qualitative and quantitative data as well as reported findings. Discrepancies were discussed together until reaching consensus. |
| Study Risk of Bias Assessment<br>Reporting Bias Assessment | The process for reducing bias in sampling. | After the abstracts were reviewed, the first author coded the studies for inclusion and all three authors worked collaboratively to verify the inclusion of each study for review. We examined the articles for their quality of 1) design, 2) evaluation, and 3) analysis and documented features of each article in tables (see S1 File). To examine bias within the articles reviewed, we documented if authors declared any conflicts of interest. |
| Effect Measures<br>Synthesis Methods<br>Certainty Assessment | The process of synthesizing and determining themes. | The first author coded the data, examining study challenges, theoretical construct, program components, evaluative components, evidence based success, evaluative improvement needed, and study bias. All three authors worked collaboratively to verify the quality of 1) design, 2) evaluation, and analysis) for each study by noting study features, theory, methods, and findings. Discrepancies were discussed together until reaching consensus. |

journals of Academic Medicine, Medical Education, Medical Teacher, and BMC Medical Education of which 18 were excluded due to not being classified by the journal as a research study containing reported findings with qualitative, quantitative, or mixed qualitative and qualitative data (see Fig 1). To be eligible for review, the articles needed to be research studies, which have explicit methods, data, and findings sections. All three authors were involved in the screening for inclusion and quality and eligibility assessment. The majority of the research studies (N = 19) are from the past ten years (see Fig 2) and the majority of research studies (N = 12) from the Journal of Academic Medicine (see Fig 3).

## Eligibility criteria for quality

Establishing the effectiveness for studies on healthcare education outreach programs required empirical evidence [27], which needed to include quantitative measures like midterm exam scores, final exam scores, concept inventory scores, etc.; and/or qualitative measures like interview data revealing student behaviors, affective factors, intention to pursue healthcare professions, etc. [28]. We categorized the report of each study on the effectiveness of the reported program based on examination criteria listed in Fig 4. For more in-depth analysis of each study on healthcare education outreach programs, see **S1A and S1B Table in S1 File** [27–31].

## Coding and analytical strategy

After the sample of (N = 37) was established, the first author coded the articles by noting the barriers, theoretical construct, program components, duration, evaluative components,

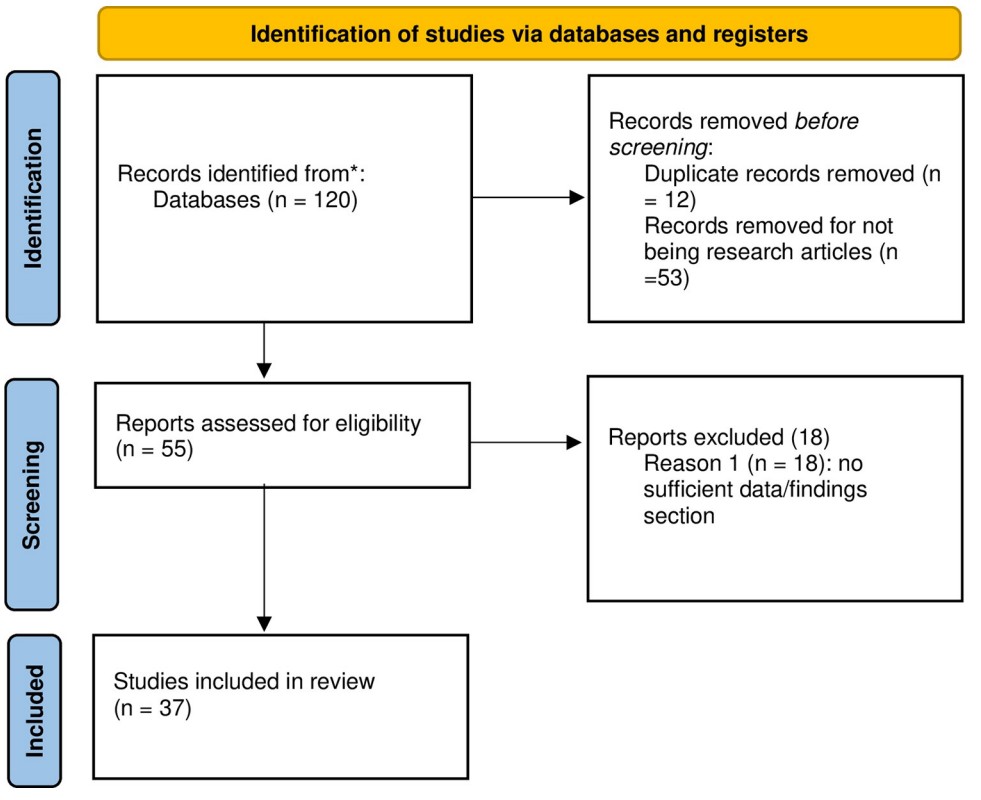

**Fig 1. Illustrates the criteria for inclusion of articles.**

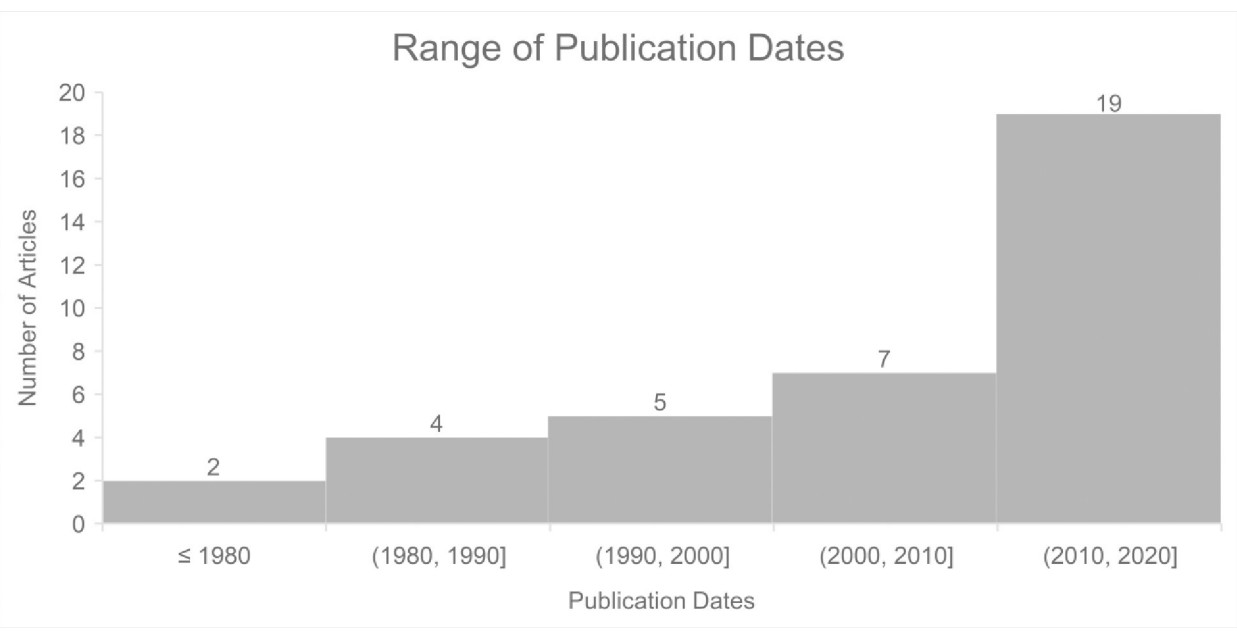

**Fig 2. Bar graph showing the range of publication dates for the selected articles in the literature review (N = 37).** There is a 10-year range per each article, with most of the articles (n = 19) being within the last ten years.

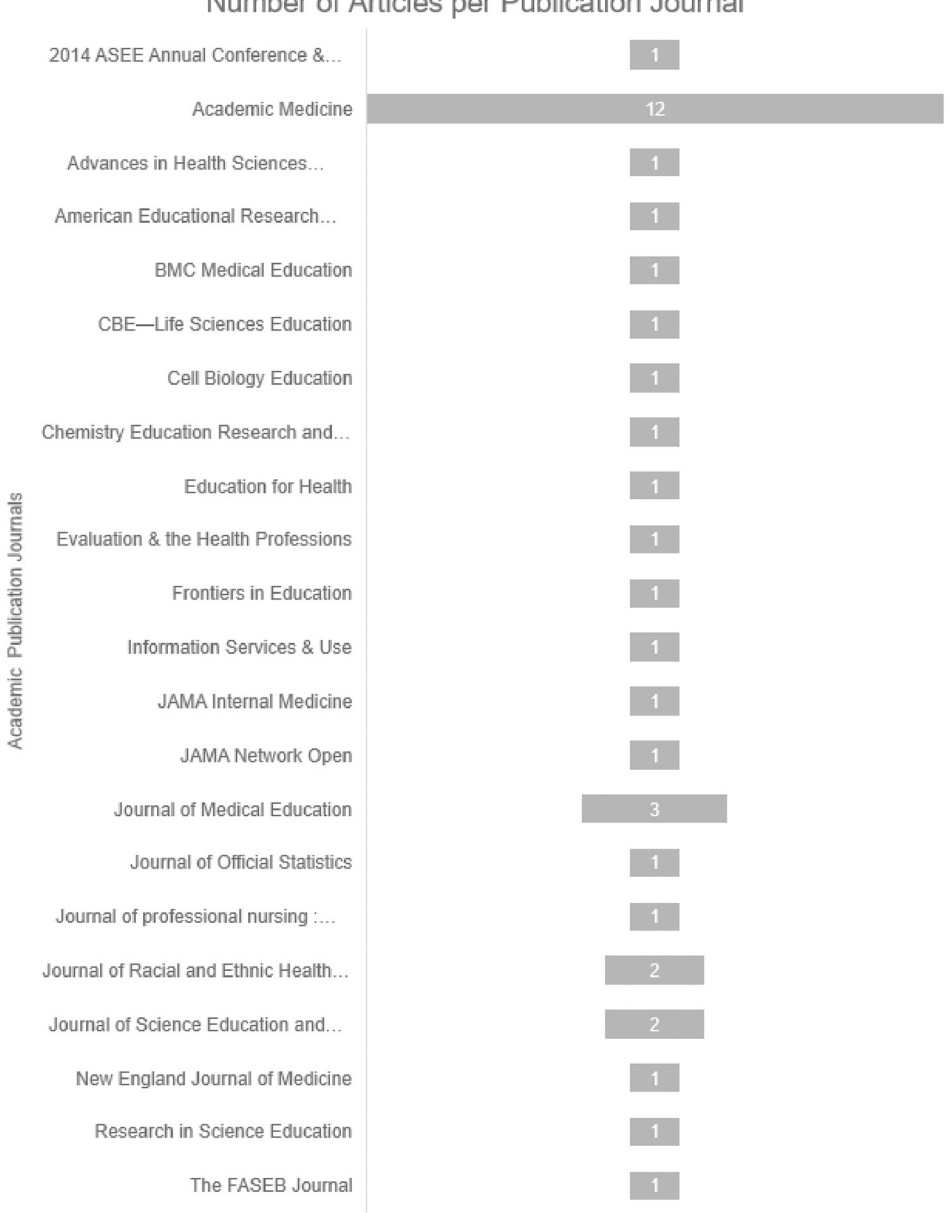

**Fig 3. Chart illustrating the number of articles per publication journal for the literature review (N = 37).** Most of the articles are from the Journal of Academic Medicine (n = 12).

evidence-based success, evaluative improvement needed and bias of study (See **S1A and S1B Table in S1 File**). The examination criteria was then organized into 1) Design, 2) Evaluation, and 3) Analysis to assess the adequacy of each study on healthcare education outreach programs. Authors worked collaboratively to determine the studies included in the systematic review had qualities for 1) design, 2) evaluation, and 3) analysis. The three criteria are crucial for any study design because they are interdependent, meaning the effects of design will carry over into evaluation and the effects of evaluation will carry over into analysis. **1) Design** refers to how studies on healthcare education outreach programs used theoretical considerations from the literature to support access for historically marginalized students. **2) Evaluation**

| Design | Evaluation | Analysis |
|---|---|---|
| • Research Question<br> • How are studies on healthcare education outreach programs designed by drawing from theory?<br><br>• Coding Features<br> • Barriers, Theoretical Construct, Program Components<br><br>• Adequate Quality Criteria<br> • Design has a present theoretical framework<br> • Design has a theoretical framework aligned with study goals | • Research Question<br> • How are studies on healthcare education outreach programs evaluated using quantitative, qualitative, and mixed-methods?<br><br>• Coding Features<br> • Duration, Evalutive Components<br><br>• Adequate Quality Criteria<br> • Data uses an appropriate comparison group<br> • Data includes both pre-measures and post-measures | • Research Question<br> • How are studies on healthcare education outreach programs analyzed to generate reliable and generalizable results?<br><br>• Coding Features<br> • Evidence Based Success, Evaluative Improvement Needed, Bias<br><br>• Adequate Quality Criteria<br> • Analysis has reported response rate higher than 80%<br> • Analysis has sufficient sample size<br> • Analysis includes qualitative data |

**Fig 4. Illustrates the criteria for the coding and analytical strategy of the articles reviewed.**

refers to how the studies on healthcare education outreach programs collected data for measurement. Finally, **3) Analysis** refers to how studies on healthcare education outreach programs verified the effectiveness for historically marginalized students pursuing future healthcare professions. The studies were determined to be adequate if the studies answered the questions in Fig 4.

## Results

Overall, the results of the systematic literature review suggest that there is a need for studies on healthcare education outreach programs to integrate empirical data with practitioner expertise to understand the unique barriers of historically marginalized students pursuing healthcare professions. Moving away from the pipeline metaphor and towards sustainable progress, studies on healthcare education outreach programs should 1) understand the main barriers to historically marginalized students entering, persisting, and succeeding in healthcare professions; 2) the protective factors that are correlated with increased entry, success, and persistence into healthcare professions; and finally, 3) the components that can successfully mitigate barriers or facilitate protective factors [18–22]. Table 2 Illustrates the percentage of studies that satisfied each category of the examination criteria. Of the N = 37 studies from the literature review, 54.2% (n = 20) met the design criteria, meaning that the studies on healthcare education outreach programs had a theoretical framework that guided design. Similarly, 54.2% (n = 20) of

**Table 2. Percentage of reviewed articles that met the selected criteria (N = 37).**

| Percentage of Programs Meeting Design Criteria | Percentage of Programs Meeting Design and Evaluation Criteria | Percentage of Programs Meeting Design Evaluation, and Analysis Criteria |
|---|---|---|
| 54.2% (n = 20) | 54.2% (n = 20) | 4.2% (n = 2) |

the studies on healthcare education outreach programs met the design and evaluation criteria, meaning studies had accurate baseline comparison (e.g., pre-measurements and post-measurements) and had an appropriate comparison group. Finally, only 4.2% (n = 2) of the studies on healthcare education programs met the design, evaluation, and analysis criteria together, meaning studies had a proper response rate and presented quantitative and/or qualitative results effectively. All articles in the review had no conflicts of interest to report, indicating low risk for bias. For full study characteristics of each article presented in the literature review, see **S1A and S1B Table in S1 File**. For each of the criteria, we will present the challenges, sub-challenges, and examples of selected studies on healthcare education outreach programs from the systematic literature review.

## 1. Challenges in design

We considered studies on healthcare education outreach programs to be adequately designed if there was an explicit theoretical framework section and the study goals could be supported by the measured outcomes. Complete theoretical framework sections contained hypotheses/research questions which drew from either literature, prior research performed, or an existing model. We considered studies on healthcare education outreach programs to be inadequately designed if there was no theoretical framework section or the study goals could not be supported by the measured outcomes. Incomplete theoretical framework sections did not contain hypotheses/research questions that drew from either literature, prior research performed, or an existing model. From the inadequately designed studies on healthcare education outreach programs, we determined two main challenges, which we have labeled as a) incomplete or missing theoretical framework; and b) misaligned study goals.

a) *Incomplete or missing theoretical framework*. From the systematic literature review, 54.2% (n = 20) of the studies on healthcare education outreach programs contained an explicitly labeled theoretical framework, hypotheses/research questions, and supported study goals. On the other hand, 45.8% (n = 17) of the studies on healthcare education outreach programs either did not have an explicit theoretical framework section, had missing hypotheses/research questions, or unsupported study goals (see Fig 5). For example, a study that tracked gifted historically marginalized students who participated in the Student Educational Enrichment Program (SEEP) through a questionnaire had no theoretical framework section to define what was meant by gifted [32]. Accordingly, the outcomes of the study explained that gifted historically marginalized students were more likely than their non-gifted counterparts to pursue healthcare professions; however, determination was predicated only on interest rather than enrollment or graduation numbers. Overall, the studies on healthcare education outreach programs without theoretical frameworks were usually from before 2010 (n = 16), indicating a trend that recently published research recognizes the value of theoretical frameworks in design (see Fig 2).

b) *Misaligned study goals*. Of the 54.2% studies on healthcare education outreach programs with detailed theoretical frameworks, 54.2% (n = 11) had study goals that aligned with the theoretical framework while 45.8% did not have aligned study goals (see Fig 6). The studies on healthcare education outreach programs with detailed theoretical frameworks had missions to provide academic and disciplinary content knowledge enrichment and mentorship through shadowing or interning with a healthcare professional (for examples see [33–36]). The structure of these healthcare education outreach programs had sustainable components such as: scholarly presentations, college immersion experiences, mentorship greater than

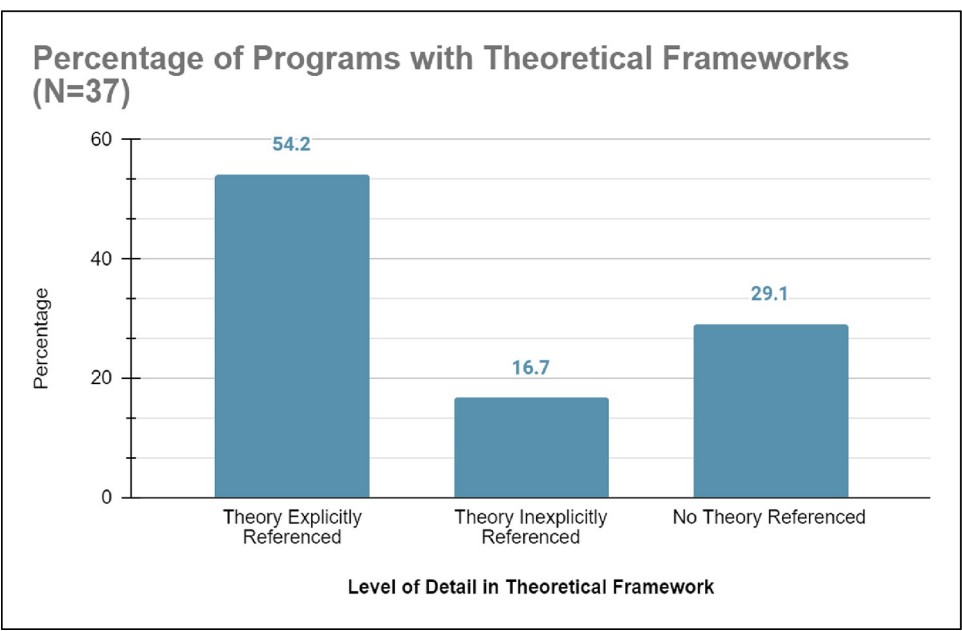

**Fig 5. Shows that 54.2% of programs studied specifically detailed a theoretical framework for the design of their intervention, 16.7% of programs studied mentioned but did not specifically detail a theoretical framework for the design of their intervention, and 29.1% of programs studied did not reference any theoretical framework.**

1-year, medical content exposure, and/or college admissions/application counseling. In contrast, unaligned studies on healthcare education outreach programs provided less sustainable components like no mentorship or had mentorship that was less than one year (See Fig 7) [37, 38] despite these studies mentioning in their literature reviews and discussions

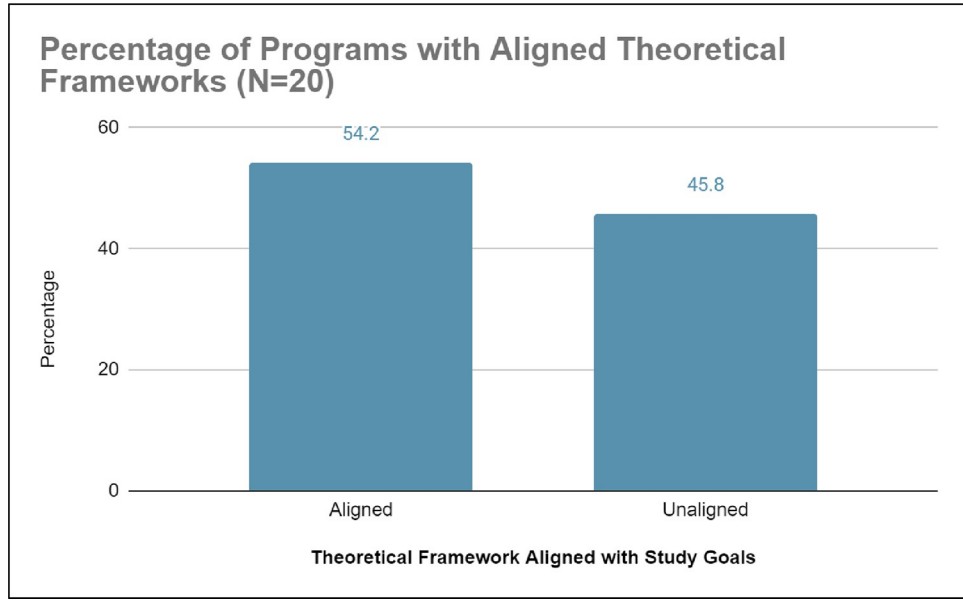

**Fig 6. Shows that 54.2% of programs studied aligned their evaluation design with their specifically detailed theoretical framework and 45.8% of programs studied either did not align their evaluation with their theoretical framework or did not reference any theoretical framework.**

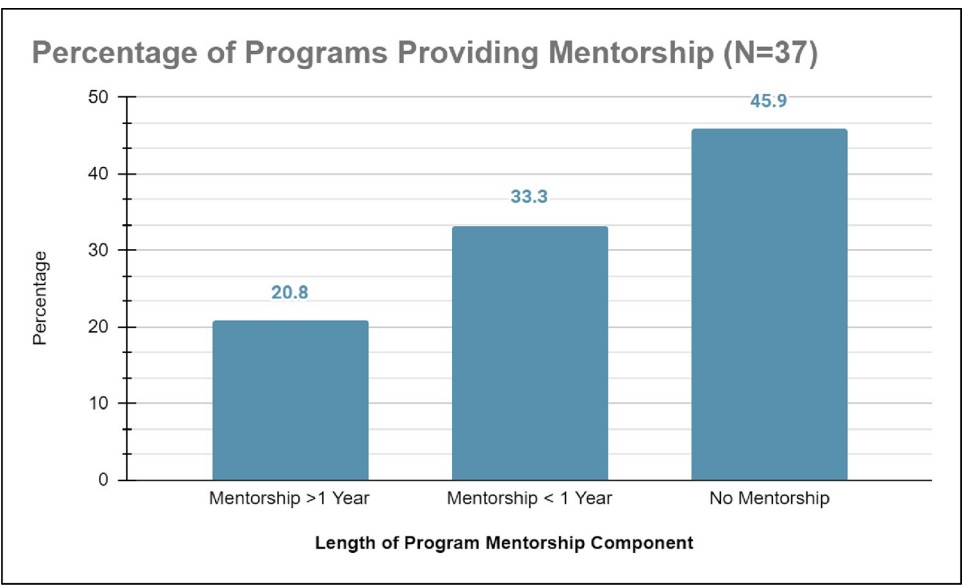

**Fig 7. Shows that 45.0% of programs studied did not include a mentorship component.** Of the programs with a mentorship program, 35.0% implementing mentorship programs of less than a year's duration, 20.0% implementing mentorship programs greater than one year's duration.

that long-term mentorship was necessary to retain historically marginalized students in STEM degrees, leading to healthcare professions [39–41].

## 2. Challenges in evaluation

We considered studies on healthcare education outreach programs to have adequate evaluation if there was an appropriate comparison group along with pre-measurements and post-measurements. The most common evaluative measurement tools among studies on healthcare education outreach programs were surveys that captured interest, awareness of healthcare careers, college enrollment, and college graduation. Appropriate comparison groups were those that closely matched key characteristics (e.g., demographic, socioeconomic, and first-generation backgrounds) of the experimental group while appropriate pre-measurements and post-measurements established a baseline to examine the efficacy of the healthcare education outreach programs. We considered studies on healthcare education outreach programs to have inadequate evaluation if there were no comparison groups, an inappropriate comparison group (i.e., key characteristics did not match) or if there were no pre-measurements and post-measurements. From the inadequately evaluated studies on healthcare education outreach programs, we determined two main challenges, which we have labeled as a) inappropriate comparison groups; and b) no pre-baseline measurements.

*a) Inappropriate comparison groups.* From our systematic literature review, 15.0% (n = 6) of studies on healthcare education outreach programs had appropriately matched comparison groups, which were generated by comparing outcomes of historically marginalized students enrolled in the healthcare education outreach programs with the outcomes of counterparts from alike schools, states, demographic backgrounds, levels of household income, and/or parent's generational status [33]. Conversely, 55.0% (n = 20) of studies on healthcare education outreach programs did not have any comparison groups while 30.0% had

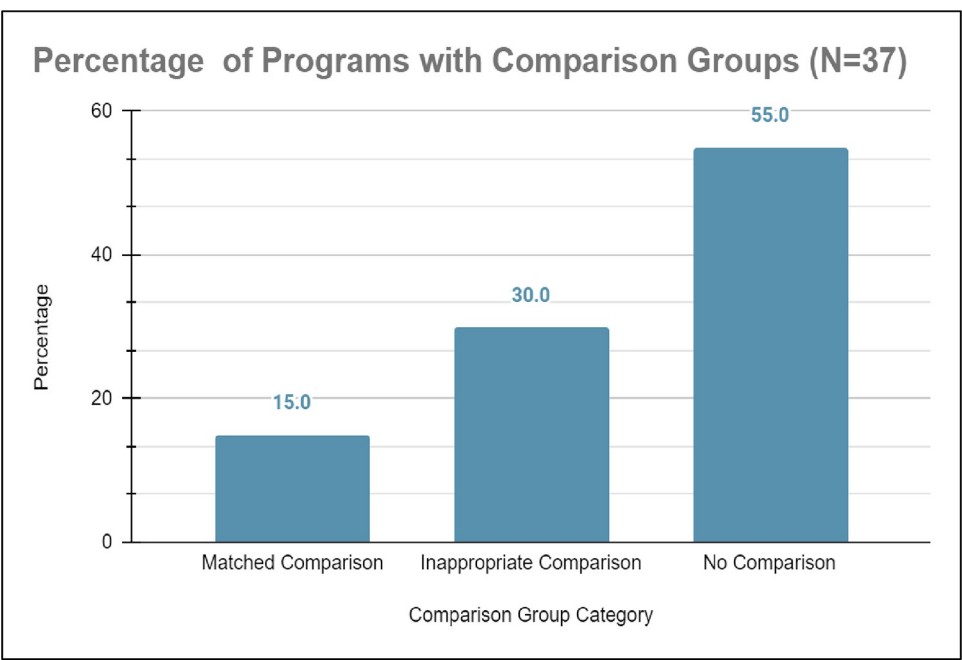

**Fig 8. Of the N = 20, only 15.0% of programs studied correctly used a matched comparison group in their study design, 30.0% of programs studied erroneously used an inappropriate comparison group in their study design, and surprisingly 55.0% of programs studied neglected to use any comparison group in their study design.**

inappropriately matched comparison groups (see Fig 8). Studies with no comparison groups had sparse data, typically using recruitment numbers into the healthcare education outreach program and enrollment into undergraduate education as the sole measure for efficacy [42, 43]). Moreover, studies with inappropriate comparison groups often 1) compared historically marginalized students to their historically non-marginalized counterparts; 2) compared historically marginalized students without matching key characteristics; or 3) compared historically marginalized students with no other group. These studies did not compare outcomes of historically marginalized students with counterparts not enrolled in the healthcare education outreach programs and often screened based on GPA, prior coursework, and academic record (for examples see [44, 45]). Unfortunately, 45.8% (n = 17) of studies also generated comparison groups using a competitive application process to select historically marginalized students who have already demonstrated evidence of science interest, efficacy, and academic success to participate in their healthcare education outreach program (see Fig 9).

b) *No pre-baseline measurements.* As shown in Fig 10, 22.7% (n = 8) of studies on healthcare education outreach programs used pre-measurements and post-measurements to evaluate efficacy. Pre-measurements were typically data collected before or within the first two weeks the healthcare education outreach program and used to create appropriate comparison groups [33]. Examples of pre-baseline measurements include data measuring content knowledge, interest, belonging, identity, or self-efficacy in pursuing healthcare professions [46, 47]. However, 68.2% (n = 25) of studies did not generate a baseline of historically marginalized students' current academic performance in and motivation of pursuing healthcare professions. Accordingly, studies on healthcare education outreach programs that only

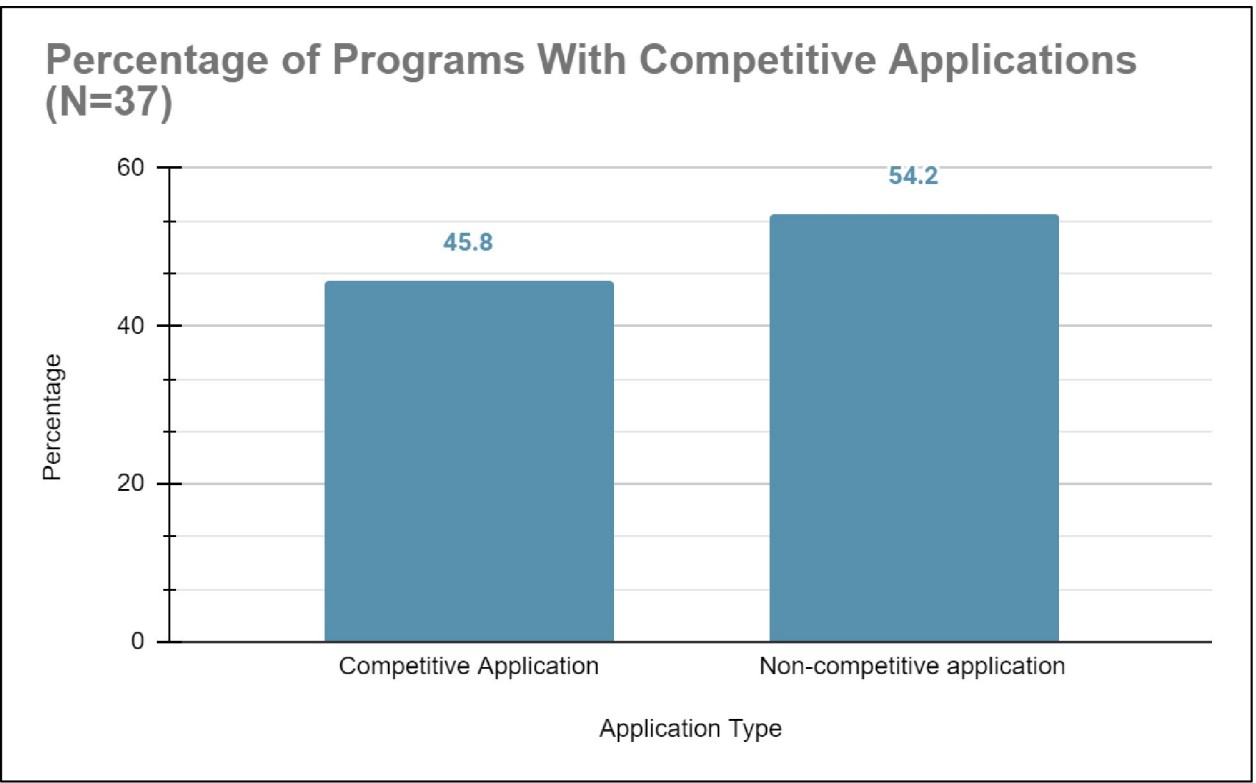

**Fig 9. Shows that 45.8% of programs studied used a competitive application process when selecting which URM and FGLI students would be allowed to participate in their healthcare pipeline program and 54.2% did not.**

collected post-measurements neglected the opportunity to compare historically marginalized students to their pre-measurement selves. Thus, there was no way to reliably evaluate how these studies on the healthcare education outreach programs supported historically marginalized students to pursue healthcare professions [37, 47, 48].

### 3. Challenges in analysis

We considered studies on healthcare education outreach programs to have adequate analysis if there was a substantial response rate and quantitative and/or qualitative analysis. Substantial response rate should be above 80% with a large enough sample size for quantitative and/or qualitative analysis [49]. We considered studies on healthcare education outreach programs to have inadequate analysis if the response rate was below 80% and if qualitative data was not used to supplement for small sample size. From the inadequately analyzed studies on healthcare education outreach programs, we determined two main challenges, which we have labeled as a) low response rate and effect size; and b) underutilization of qualitative methods.

*a) Low response rate and effect size.* From our systematic literature review, 37.5% (n = 14) had a reported response rate above 80%, meaning that of the data collected, at least 80% of participants' data was included in the analysis (example [50]). Studies with high response rates disaggregated by historically marginalized group and had generalizable outcomes such as the mean performance of historically marginalized students that afforded enrollment into undergraduate education and eventually graduation. In contrast, 40.0% (n = 15) of studies

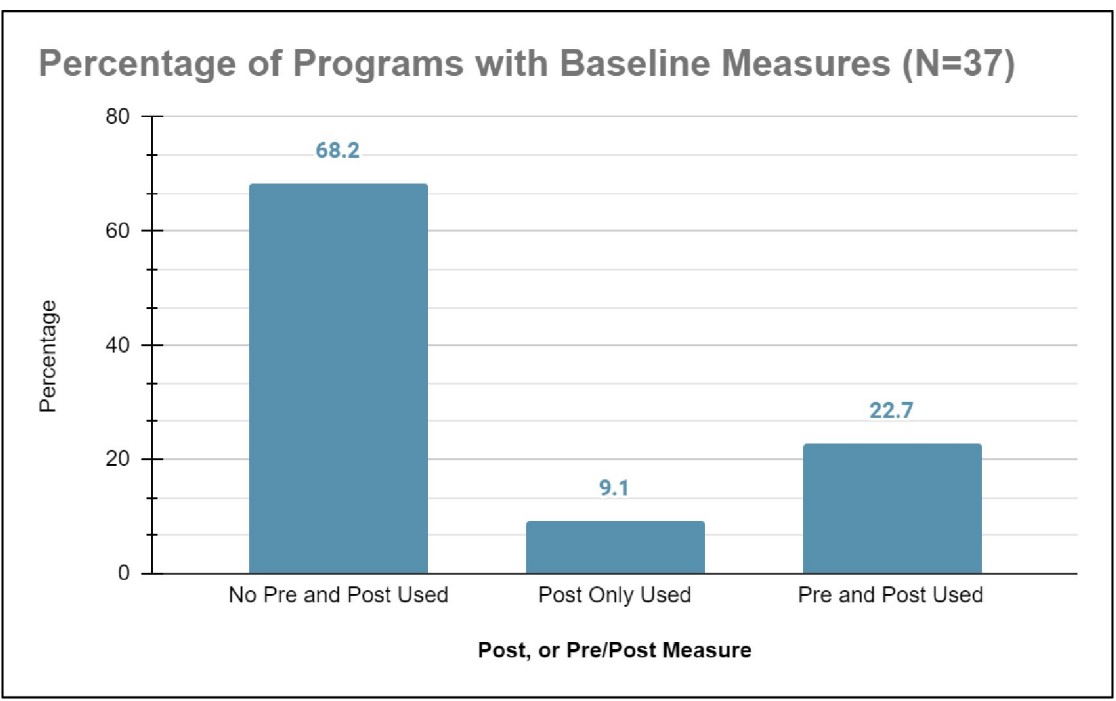

**Fig 10. 22.7% of programs studied correctly used both pre-program and post-program measures, 9.1% used post-only measurements, and surprisingly 68.2% neglected to use either pre or post-program measures.**

had response rates of less than 60.0% and tended to have competitive application processes [51]. These studies had small sample sizes that could not determine effect size, statistically significant components, or generalizable results (example [33, 37, 51]). Finally, 29.2% (n = 11) of the data did not report any response rate (see Fig 11).

b) *Underutilization of qualitative methods.* Fig 12 shows 24.0% (n = 9) of studies on healthcare education outreach programs utilized qualitative data in the analysis. Examples of qualitative data included student interviews [36, 45, 52]) and open-response text from surveys (example [53]). The role of the qualitative data was to triangulate with the quantitative findings (example [36]) to supplement for small sample size. In our systematic literature review, there were no studies on healthcare education outreach programs that used only qualitative data. However, 76.0% (n = 28) of studies on healthcare education outreach programs utilized only quantitative data analysis, which included a variety of parametric and non-parametric tests.

## Limitations, discussion, and implications

The systematic literature review contains the following limitations that impact how these results should be interpreted: 1) the number of reviewed articles (N = 37) was too small for an in-depth quantitative analysis; 2) we only included research with explicit methods, data, and findings sections, which may have obscured the number of studies that contained only descriptions of research efforts; 3) we used the declaration of conflicts of interest in each article to measure bias, which is not an encompassing approach [54]. Nevertheless, we believe despite these limitations, the findings generate important discussions surrounding the efficacy of

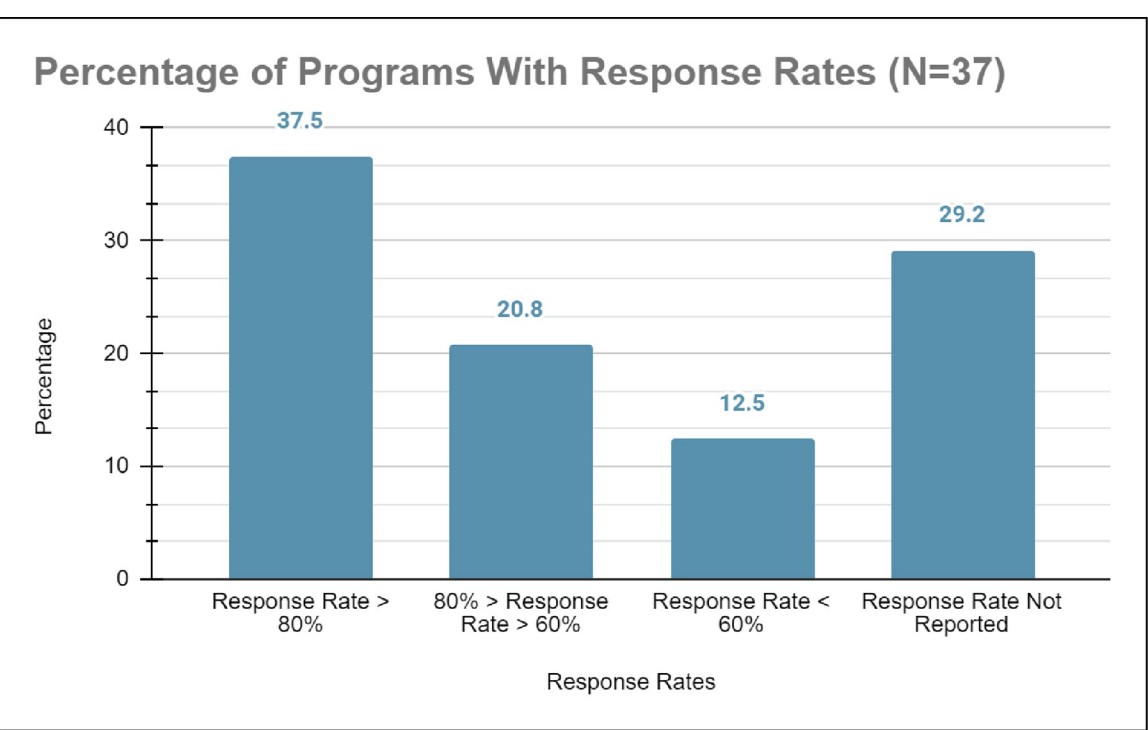

**Fig 11. Shows that 37.5% of programs studied had response rates above 80%; 20.8% of programs had response rates between 60% and 80%; 20.8% had a response rate below 60% and 29.2% had no reported response rate.**

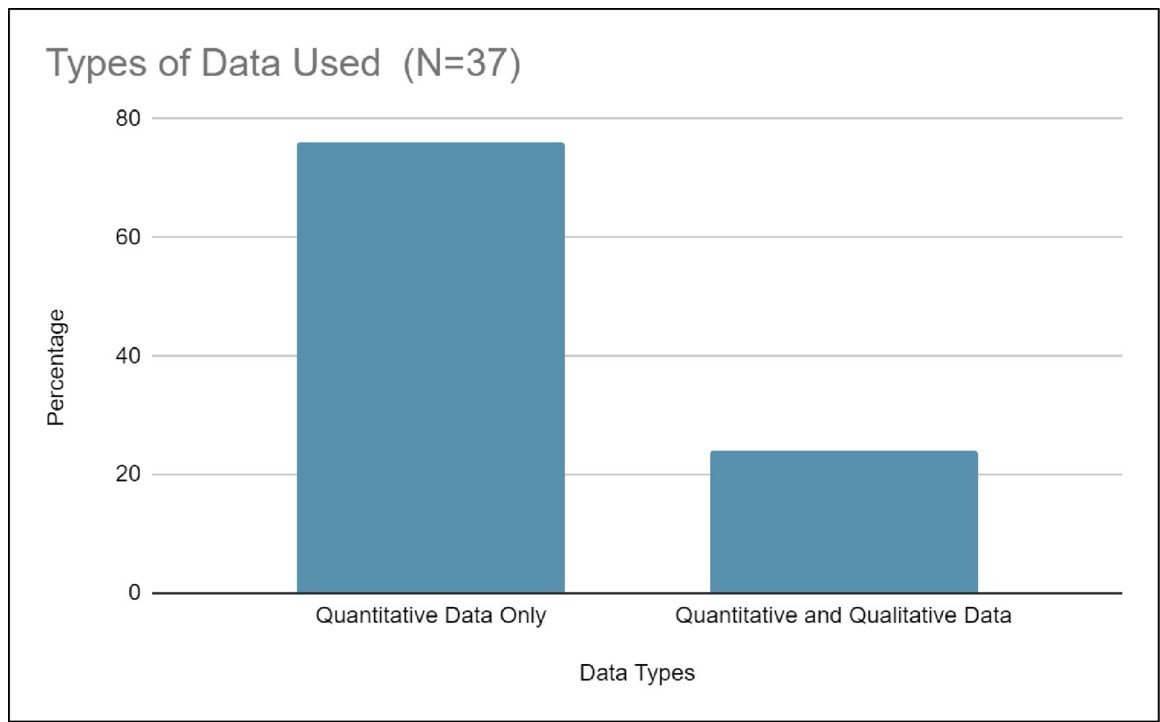

**Fig 12. Shows that 24.0% of the studies on outreach programs utilized a combination of qualitative and quantitative data together, whereas the majority at 76.0% only utilized quantitative methods.**

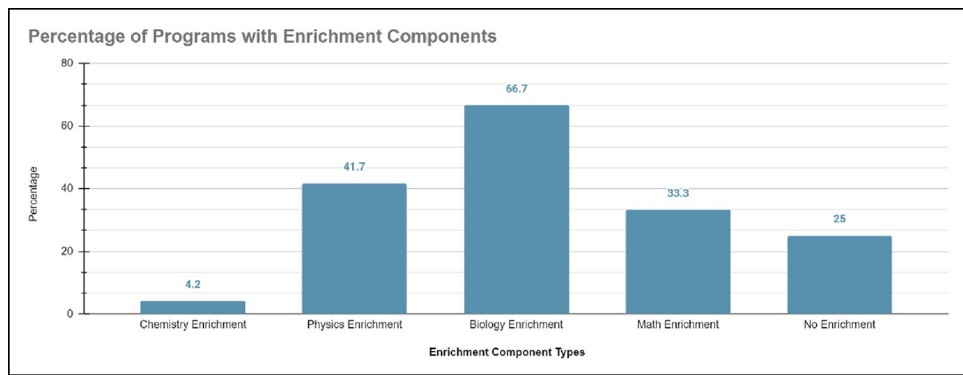

**Fig 13. Shows that 66.7% of programs had biology enrichment, 41.7% had physics enrichment, 33.3% had math enrichment, and 4.2% had chemistry enrichment.** 25.0% of studies had no content enrichment.

healthcare programs for historically marginalized students. The following section will elaborate on why design, evaluation, and analysis are integral features to study efficacy of outreach programs by drawing on related literature in cognitive psychology, science education, and discipline-based education research. To elaborate why these features are effective, we will also provide examples on how to improve the quality of research. Fig 14 below illustrates study features and highlight components for effective studies on healthcare education outreach programs that researchers can use to support their practice.

Beginning with design, most of the studies on healthcare education outreach programs did not have theoretical frameworks. Lack of theory in study design is a major challenge because theoretical frameworks help researchers guide hypotheses [55]. Through theory-driven hypotheses, causal mechanisms can be generated that operationalize variables of interest with

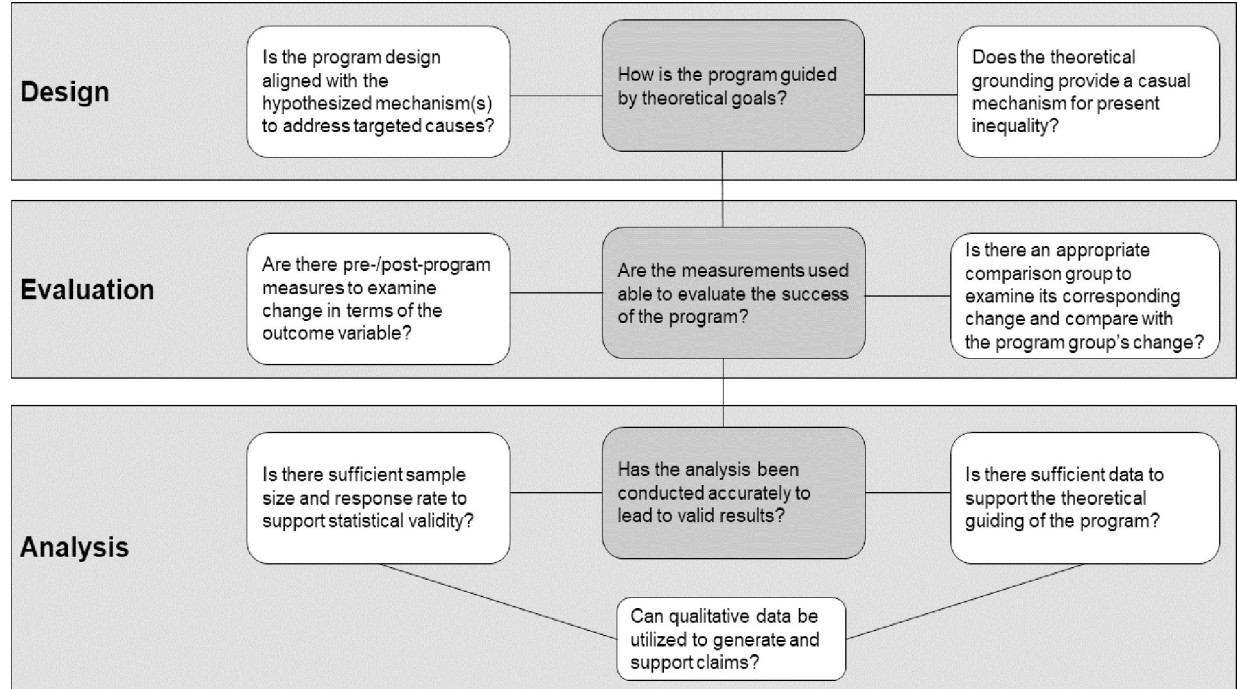

**Fig 14. Illustrates the connection among design, evaluation, and analysis features.**

the expected outcomes, which would have helped some studies on healthcare education outreach programs to better structure program components like mentorship and content learning [8]. Without theoretically-driven hypotheses, studies on healthcare education outreach programs are less likely to have an effective design that can be properly tested by selecting previously validated instruments and/or methods from the literature. The focus on theoretical frameworks and resulting hypotheses is also crucial for replicating structures across studies on healthcare education outreach programs. Explicitly using and naming theoretical frameworks has been shown to develop interdisciplinary research [56], which can address complex problems like healthcare disparity in the United States through reliable studies on healthcare education outreach programs that support historically marginalized students to pursue healthcare professions.

Within our literature review, studies on healthcare education outreach programs that missed the design criteria often lacked a mentorship component (e.g., college preparation counseling curriculum or advising), which has been shown to be a protective factor in supporting historically marginalized students pursuing STEM degrees and eventually healthcare professions [55–57]. Mentorship often provides historically marginalized students with social capital, which is a sociological term from Pierre Bourdieu that shows how access to knowledges, resources, and experiences can better facilitate social navigation [57]. Essentially, mentorship provides historically marginalized students access to knowledge about selecting a college, financial aid information/scholarships, and knowing how to enroll when accepted, which is more challenging for students who are first-generation scholars or a person from a low socioeconomic background without existing family members in academia and in healthcare professions [56, 57]. In contrast, studies on healthcare education programs that met the design criteria not only had a mentorship component for historically marginalized students, but also for the families [39]. For example, college-advising workshops and parent-support workshops were shown to help historically marginalized students and their families understand expected coursework, financial aid opportunities for tuition and supplies, as well as on-campus tutoring and learning centers [39].

Analogously, studies on healthcare education outreach programs that met the design criteria also heavily emphasized teaching historically marginalized students the disciplinary content knowledge needed to be successful in STEM majors and therefore in healthcare professions. Our systematic literature reveals 25.0% (n = 9) of studies on healthcare education outreach programs did not have disciplinary content knowledge enrichment, while 75% (n = 28) had disciplinary content knowledge enrichment. Of the studies with disciplinary content knowledge enrichment, 66.7% (n = 19) of studies had biology enrichment, 41.7% (n = 12) had physics enrichment, 33.3% (n = 9) had math enrichment, and 4.2% (n = 1) had chemistry enrichment (see Fig 13); studies on healthcare education programs had multiple enrichments, meaning percentages will be over 100%. Previous works have shown that preparation gaps exist between historically marginalized and historically non-marginalized students [58] and that introductory chemistry serves as a major gatekeeper for historically marginalized students to pursue STEM degrees and eventually healthcare professions [58]. The documented preparation gaps in mathematics and science courses like biology, chemistry, and physics further highlight the education disparity for historically marginalized students compared to their counterparts. Thus, studies on healthcare education outreach programs that emphasized disciplinary content knowledge enrichment, especially for chemistry, illustrate how the use of theory can guide structures that best serve historically marginalized students pursuing healthcare professions.

In terms of the evaluation criteria, most studies on healthcare education outreach programs had challenges evaluating differences between historically marginalized and historically non-

marginalized students. As mentioned previously, historically marginalized students face disproportionate education disparity when compared to their counterparts [59]; consequently, academic preparedness is a salient feature to consider for structuring healthcare education outreach programs. However, academic preparedness in studies on healthcare education outreach programs often used academic preparedness to discriminate applicants, favoring historically marginalized students who have already demonstrated academic proficiency and an inclination towards pursuing STEM degrees [60]. Instead, some studies on healthcare education outreach programs have accepted students into the program if historically marginalize students had low academic preparedness (i.e., GPA around 3.0) and had a neutral to slightly positive inclination towards pursuing healthcare professions [38, 50]. To address challenges in evaluating differences between historically marginalized and historically non-marginalized students, some studies on healthcare education outreach programs have suggested matching applicants based on one or two similar criteria such as underrepresented minority status, household income level, parental highest education achievement, and baseline GPA rather than evaluating program success in aggregate [32, 50].

Finally, the analysis criteria revealed most studies on healthcare education outreach programs needed either larger sample sizes or longitudinal data to verify claims that the healthcare education outreach program had long-term effects on historically marginalized students pursuing STEM majors and healthcare professions. When analyzing self-reported measurements, response rate is critical because small sample sizes can significantly decrease validity and generalizability of findings [60]. In quantitative analysis, the study outcomes begin to lose validity when response rates fall below 60% due to the uncertainty that the sample of respondents accurately reflects elements of the complete population of interest [61]. To address low response rates, researchers should ask whose responses are missing and what nonresponse bias means for the study goals. For historically marginalized students, nonresponse bias can reveal that an aspect of the healthcare education outreach program is inaccessible (e.g., requiring high academic preparation) [38, 50]. Researchers should also provide information about the sampling process employed, response rates calculations, as well as utilize qualitative research methods to strengthen the evidence being reported, particularly when it challenges the sample size required for quantitative analyses.

Furthermore, studies on healthcare education outreach programs were understandably constrained financially since they were self-supported through grant or university allocation funding. Thus, studies lacked the funding to achieve long-term follow up on whether historically marginalized students in the healthcare education outreach programs went on to pursue healthcare professions. To address challenges with acquiring long-term follow up with historically marginalized students, studies have used the completion of college pre-med freshman and sophomore science coursework to serve as sufficient predictors of persistence and future STEM major graduation [62]. Multiple modes of contact, follow-up appearance, incentives, personalization, and longitudinal mentorship were also used, which had significant impacts on survey response rates after the healthcare outreach program was completed [49]. Researchers can also combine quantitative pre- and post-program measures with qualitative research methods using an appropriate theoretical framework to support the reliability of the analysis [63].

Overall, we strongly suggest researchers invested in reducing disparities of racial and ethnic groups in the healthcare field utilize a guiding theoretical framework to design studies on healthcare education outreach programs and constructs of interest, which will then support proper evaluation and analyses. For proper evaluation, it is important to make sure there is an appropriate comparative group and pre-and post-measures to measure the efficacy of the program. To analyze the efficacy, researchers should make sure their results are reliable and

generalizable by ensuring representative sample size for quantitative analysis and/or triangulating quantitative analysis with in-depth qualitative analysis. Fig 14 shows a detailed flow chart of decisions researchers should consider when structuring healthcare education outreach programs.

## Conclusion

To support historically marginalized students' pursuit of and persistence in healthcare professions, studies on healthcare education outreach programs should structure components that foster sustainability through mentorship, financial support, and disciplinary content knowledge enrichment. It is crucial that early efforts to structure healthcare education outreach programs deliberately address equity barriers that historically marginalized students disproportionately face in pursuing healthcare professions through the education system. Deliberately addressing equity barriers for historically marginalized students includes doing studies with 1) a strong theoretical foundation with aligned study goals; 2) an appropriate comparison group with measurements taken before, during, and after the healthcare education outreach program; and 3) a reliable method to ensure the outcomes are replicable and generalizable. Ultimately, addressing challenges in design, evaluation, and analysis can better support future researchers interested in studying healthcare education outreach programs to structure them successfully for historically marginalized students to pursue STEM degrees, leading them eventually to pursue high-paying careers in healthcare professions. The advancement of historically marginalized students is a critical step towards achieving equity and social justice for historically marginalized communities and as healthcare professionals, historically marginalized students would therefore be important frontier agents to mitigate healthcare disparity in the United States.

## Supporting information

**S1 Checklist.**
(DOCX)

**S1 File.** S1A Table: Barriers, assumptions, components, evaluative methods, conclusions and evidence-based success. S1B Table: Recommendations, reasoning, and conclusions of non-program high school healthcare education pipeline research.
(DOCX)

## Author Contributions

**Conceptualization:** Eric Jenkins, Jocelyn Elizabeth Nardo, Shima Salehi.

**Data curation:** Eric Jenkins.

**Formal analysis:** Eric Jenkins, Shima Salehi.

**Investigation:** Eric Jenkins, Shima Salehi.

**Methodology:** Jocelyn Elizabeth Nardo, Shima Salehi.

**Project administration:** Shima Salehi.

**Supervision:** Shima Salehi.

**Validation:** Shima Salehi.

**Writing – original draft:** Eric Jenkins, Jocelyn Elizabeth Nardo.

**Writing – review & editing:** Jocelyn Elizabeth Nardo, Shima Salehi.

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
