## [Decision Letter · Decision Letter 0]

14 Sep 2022

PONE-D-22-13001A Systematic Review of the 60 Year Literature: Effects of Outreach Programs in Supporting Historically Marginalized and First-Generation, Low-Income Students in Healthcare EducationPLOS ONE

Dear Dr. Nardo,

Thank you for submitting your manuscript to PLOS ONE. After careful consideration, we feel that it has merit but does not fully meet PLOS ONE’s publication criteria as it currently stands. Therefore, we invite you to submit a revised version of the manuscript that addresses the points raised during the review process.

I agree with the two reviewers that your paper focuses on an important topic in the field. However, I also found valuable insights from reviewers about the issues of this paper including the language and organization of this paper and details about the study design. First, I think the organization of this paper is more like a report rather than a publishable manuscript. Please follow PLOS ONE submission guidelines. Also, please polish and proofread the language. Second, provide more details of the included papers for each evaluation criteria. For programs that have theoretical frameworks, what are their frameworks? For misaligned study goals, how are they misaligned? What about those having aligned study goals? In other words, please provide more details than descriptive summaries. Third, how and why the coding strategy was developed? Is it following any theoretical framework? 

We look forward to receiving your revised manuscript.

Kind regards,

Xiaodan Tang

Academic Editor

PLOS ONE

Journal Requirements:

“NO”

3. Please ensure that you refer to Figure 1-3 in your text as, if accepted, production will need this reference to link the reader to the figure.

Reviewers' comments:

Reviewer's Responses to Questions

**Comments to the Author**

1. Is the manuscript technically sound, and do the data support the conclusions?

Reviewer #1: Yes

Reviewer #2: Yes

2. Has the statistical analysis been performed appropriately and rigorously? 

Reviewer #1: N/A

Reviewer #2: N/A

3. Have the authors made all data underlying the findings in their manuscript fully available?

Reviewer #1: Yes

Reviewer #2: Yes

4. Is the manuscript presented in an intelligible fashion and written in standard English?

Reviewer #1: No

Reviewer #2: Yes

5. Review Comments to the Author

Reviewer #1: This manuscript addresses the important issue of high school outreach programs in healthcare education. The authors identify a number of issues related to the design, analysis, and evaluation of such studies that are currently hindering progress in understanding the efficacy of such programs.

My primary concern is with the written English of the manuscript. There are many places throughout the manuscript where the sentence structure, subject-verb agreement and use of prepositions confuse the meaning of the sentences. Overall, this has a substantial impact on the points the authors wish to communicate.

Some examples:

Page 1, "non-historically marginalized" should be historically non-marginalized

Page 3, "If studies were funding/declared interest"

Page 15, "Without a theoretical driven hypothesis, first, outreach programs...."

Page 16 + 17 there are several sentences which should be carefully reviewed.

There are some additional points that I think need to be addressed which I will put in chronological order.

1. The authors do not adequately draw on the literature to establish the link between underrepresentation in healthcare and gaps in quality care outcomes for marginalized groups. I think more needs to be done to explain this.

2. The UMMC study should be described in greater detail because the primary purpose of the manuscript is to critically evaluate the existing literature. The reader needs to be convinced that the results of this study are believable given all of the shortcomings that are identified in the body of the manuscript.

3. Table 1: Study Risk of Bias Assessment: the explanation of how this was done is not at all clear to me. I cannot distinguish between this explanation and the explanation for data items, quality, and eligibility assessment.

4. (Major) I think more needs to be said about the fact that "lack of study results" was a criterion for excluding articles from analysis. Do you mean quantitative results? I think it is an important oversight that you potentially missed qualitative results in a number of the articles that the search returned. There are only 120 total articles, so I think some review should be done to ensure that you did not miss qualitative results in the excluded articles.

5. Results section, first sentence. I think you need to justify this claim with data or literature.

6. (Minor) Some figures are reported to three decimal places and others to two. Please be consistent.

7. You never explain the concept of social capital, but draw on it several times in the manuscript to justify claims. Please provide some context.

8. Page 16 "Trends in the performance outcomes" -- if these are gaps in outcomes, please say so. It is not clear form this sentence that the outcomes are undesirable.

9. Page 18: what does it mean for qualitative results to "challenges the power" required for quantitative analyses?

Reviewer #2: -Summary. The manuscript involved synthesizing extant literature related to outreach efforts targeted at supporting historically marginalized groups in pursuing healthcare fields. Overall, the systematic review was organized and well-written, with a major strength of the paper being the transparency related to clearly outlining how the data were collected. The primary suggestions for improvement are related to the discussion of data analysis and the reporting of the findings. I have some additional comments and suggestions below.

-Abstract. This provides a nice summary of the themes, previewing the organization of the paper. A minor suggestion would be to include the final sample of papers in the abstract.

-Introduction. The introduction demonstrates a strong grounding in prior literature and familiarity with current trends, including moving away from the pipeline metaphor to addressing how we can support students (as opposed to simply sending more people through the pipeline). Although there is variation in systematic reviews regarding their inclusion, a guiding research question (or questions) could be helpful to more clearly articulate the focus of the review. Based on the analysis and framing, it is clear the authors have a specific goal with reviewing the literature (i.e., evaluating overall quality of research design), which could be communicated through research question(s).

-Methods. I appreciated the transparency related to how the papers were selected and collected, as well as the table outlining how this paper meets the criteria for a systematic review paper (including the PRIMSA checklist in the supplemental). One area for improvement would be the ways in which the data analysis is discussed. Data analysis is briefly discussed in the paper, with a supplemental document providing more insight into the nature of the analysis (this supplemental paper was very helpful). After reading the paper and the supplemental, it is not clear what the coding process looked like. Would you consider each acronym in the supplemental to be a code (e.g., NO, ASP, UBP, HSC, etc.) or were the codes larger in grain size (bullet points under the criteria in Figure 4)? There are also other themes that might suggest additional codes (e.g., Figure 5 – “specifically detailed”, “not specifically detailed” – but this wording is different from the wording in the supplemental under “Theoretical Underpinning Key”, so it is unclear if they are distinct or if they map onto one another; other examples would be the other descriptive trends in the other figures like Figure 9, etc.). Also, was it one author coding or multiple authors coding and did they calculate any measures related to reliability for the code assignments?

-Findings. Broadly, the Findings are organized and have a logical flow, reflecting the previously discussed emphasis of design, evaluation, analysis. The authors mention the paper is descriptive, which is not necessarily an issue; this is common among literature reviews because you are compiling a list that summarizes the research. However, I have some suggestions about the presentation of the descriptive findings. Due to the small sample size, I would suggest moving away from quantifying the findings, particularly, reconsider reporting percentages. The percentages can be misleading because they distance the findings from the context. This is particularly the case when there is discussion about percentages involving a subset of the sample (e.g., “Of the 54.2% of studies with detailed theoretical frameworks, 54.2% had study goals that aligned with the framework”). For transparency and clarity, provide raw values instead of percentages (“n=20 of the studies had a detailed framework, with only n=11 of these studies having goals that aligned with the framework”). This suggestion applies across the findings, including reporting values in figures, tables, etc.

-Conclusion. The claims made and conclusions are appropriate and thoughtful and provide great suggestions for future work in this area. The suggestions would improve future work and afford more substantiative claims with more robust study designs and theoretical grounding.

6. PLOS authors have the option to publish the peer review history of their article (what does this mean?). If published, this will include your full peer review and any attached files.

Reviewer #1: No

Reviewer #2: **Yes: **Jon-Marc G. Rodriguez

---

## [Author Response · Author response to Decision Letter 0]

24 Oct 2022

Dear editor and reviewers, 

We greatly appreciate the careful consideration taken to improve the presentation and content of the manuscript. We believe the editor and reviewers’ comments were essential to strengthen the overall presentation and scholarship of the work. 

Reviewer 1 (see uploaded letter for full table of responses)

My primary concern is with the written English of the manuscript. There are many places throughout the manuscript where the sentence structure, subject-verb agreement and use of prepositions confuse the meaning of the sentences. Overall, this has a substantial impact on the points the authors wish to communicate.

We agree and sincerely apologize for the written English of the manuscript upon submission. We have significantly re-written every section for clarity. We have also consistently used the same verbiage (e.g., studies on healthcare education outreach programs) when explaining the same phenomenon in the text.

The authors do not adequately draw on the literature to establish the link between underrepresentation in healthcare and gaps in quality care outcomes for marginalized groups. I think more needs to be done to explain this.

We agree and the change has been made in the main document (Page 2):

The UMMC study should be described in greater detail because the primary purpose of the manuscript is to critically evaluate the existing literature. The reader needs to be convinced that the results of this study are believable given all of the shortcomings that are identified in the body of the manuscript.

-“The study at UMMC drew from the Mississippi Science Framework that theorized program components such as student research mentorship, teacher professional development, disciplinary content knowledge enrichment, and practical experiences, which has promoted sustainability of the program for 28 years(22).”

Study Risk of Bias Assessment: the explanation of how this was done is not at all clear to me. I cannot distinguish between this explanation and the explanation for data items, quality, and eligibility assessment.

We agree and the change has been made in the main document (Page 4). We also cite (Page 16) how the review of bias by seeing if studies had reported any conflicts of interest is a limitation of the work because it is not an encompassing approach: 

“After the abstracts were reviewed, the first author coded the studies for inclusion and all three authors worked collaboratively to verify the inclusion of each study for review. We examined the articles for their quality of 1) design, 2) evaluation, and 3) analysis and documented features of each article in tables (see supplemental information). To examine bias within the articles reviewed, we documented if authors declared any conflicts of interest.”

I think more needs to be said about the fact that "lack of study results" was a criterion for excluding articles from analysis. Do you mean quantitative results? I think it is an important oversight that you potentially missed qualitative results in a number of the articles that the search returned. There are only 120 total articles, so I think some review should be done to ensure that you did not miss qualitative results in the excluded articles.

We agree and for clarity, the studies needed to be classified as research studies by the journal (e.g., not editorials or tutorial papers) with defined methods, data, and findings sections. Studies needed to define the data collected and how the data was analyzed rather than a summary of what the components of the healthcare education outreach program did, which was the case for the papers excluded. We also cite this is a limitation of the work (Page 16). The change has been made in the main document (Page 4-5):

“Our second search returned 55 relevant articles from the journals of Academic Medicine, Medical Education, Medical Teacher, and BMC Medical Education of which 18 were excluded due not being classified by the journal as a research article containing reported findings with qualitative, quantitative, or mixed qualitative and qualitative data (see Fig 1). To be eligible for review, the articles needed to be research studies, which have explicit methods, data, and findings sections.”

You never explain the concept of social capital, but draw on it several times in the manuscript to justify claims. Please provide some context.

We agree and the change has been made in the main document (Page 17):

“Mentorship often provides historically marginalized students with social capital, which is a sociological term from Pierre Bourdieu that shows how access to knowledges, resources, and experiences can better facilitate social navigation(57).”

"Trends in the performance outcomes" -- if these are gaps in outcomes, please say so. It is not clear form this sentence that the outcomes are undesirable.

We agree and have removed the phrasing from the main document.

what does it mean for qualitative results to "challenges the power" required for quantitative analyses?

We agree and have removed the phrasing from the main document.

Reviewer 2 (see uploaded letter for full table of responses)

Although there is variation in systematic reviews regarding their inclusion, a guiding research question (or questions) could be helpful to more clearly articulate the focus of the review. Based on the analysis and framing, it is clear the authors have a specific goal with reviewing the literature (i.e., evaluating overall quality of research design), which could be communicated through research question(s).

We agree and the change has been made in the main document (Page 3):

Our systematic literature review seeks to answer:

How are studies on healthcare education outreach programs designed by drawing from theory? 

How are studies on healthcare education outreach programs evaluated using quantitative, qualitative, and mixed-methods?

How are studies on healthcare education outreach programs analyzed to generate reliable and generalizable results? 

One area for improvement would be the ways in which the data analysis is discussed. Data analysis is briefly discussed in the paper, with a supplemental document providing more insight into the nature of the analysis (this supplemental paper was very helpful). After reading the paper and the supplemental, it is not clear what the coding process looked like. Would you consider each acronym in the supplemental to be a code (e.g., NO, ASP, UBP, HSC, etc.) or were the codes larger in grain size (bullet points under the criteria in Figure 4)? There are also other themes that might suggest additional codes (e.g., Figure 5 – “specifically detailed”, “not specifically detailed” – but this wording is different from the wording in the supplemental under “Theoretical Underpinning Key”, so it is unclear if they are distinct or if they map onto one another; other examples would be the other descriptive trends in the other figures like Figure 9, etc.).

We agree and for clarity, we do consider each acronym to be a sub-code and they map onto the larger scheme for design, evaluation, and analysis. For the main text, we will only include the larger code category and how it aligns with the design, evaluation, and analysis to streamline the paper. We also changed the figure heading to match better what is in the Supplemental and the paper. The changes have been made in Figure 4 where we have the research question, coding features, and adequate quality criteria. 

Also, was it one author coding or multiple authors coding and did they calculate any measures related to reliability for the code assignments?

We agree and for clarity, the first author listed coded through the literature and the second and third author listed verified the codes with the first author. Disagreements were discussed together until reaching consensus. The change has been made in the main document (Page 3-5) in Table 1 and Study Search, Retrieval, Eligibility, and Selection. 

Due to the small sample size, I would suggest moving away from quantifying the findings, particularly, reconsider reporting percentages. The percentages can be misleading because they distance the findings from the context. This is particularly the case when there is discussion about percentages involving a subset of the sample (e.g., “Of the 54.2% of studies with detailed theoretical frameworks, 54.2% had study goals that aligned with the framework”). For transparency and clarity, provide raw values instead of percentages (“n=20 of the studies had a detailed framework, with only n=11 of these studies having goals that aligned with the framework”). This suggestion applies across the findings, including reporting values in figures, tables, etc.

We agree and have also decided to include the number of articles in each of the titles so that it is transparent for the reader to calculate.

---

## [Decision Letter · Decision Letter 1]

17 Nov 2022

A Systematic Review of the 60 Year Literature: Effects of Outreach Programs in Supporting Historically Marginalized and First-Generation, Low-Income Students in Healthcare Education

PONE-D-22-13001R1

Dear Dr. Nardo,

We’re pleased to inform you that your manuscript has been judged scientifically suitable for publication and will be formally accepted for publication once it meets all outstanding technical requirements.

Kind regards,

Nabeel Al-Yateem, PhD

Academic Editor

PLOS ONE

Additional Editor Comments (optional):

Reviewers' comments:

Reviewer's Responses to Questions

**Comments to the Author**

1. If the authors have adequately addressed your comments raised in a previous round of review and you feel that this manuscript is now acceptable for publication, you may indicate that here to bypass the “Comments to the Author” section, enter your conflict of interest statement in the “Confidential to Editor” section, and submit your "Accept" recommendation.

Reviewer #1: All comments have been addressed

Reviewer #2: All comments have been addressed

2. Is the manuscript technically sound, and do the data support the conclusions?

Reviewer #1: Yes

Reviewer #2: Yes

3. Has the statistical analysis been performed appropriately and rigorously? 

Reviewer #1: N/A

Reviewer #2: Yes

4. Have the authors made all data underlying the findings in their manuscript fully available?

Reviewer #1: Yes

Reviewer #2: Yes

5. Is the manuscript presented in an intelligible fashion and written in standard English?

Reviewer #1: Yes

Reviewer #2: Yes

6. Review Comments to the Author

Reviewer #1: Thank you for addressing my concerns. I believe that the revised manuscript is much stronger and more clearly written.

Reviewer #2: The authors present a strong contribution to the literature. Comments I raised such as having research questions and modifying the framing of the data analysis/findings were addressed. I especially liked how the research questions mapped onto the three qualities (design, evaluation, analysis) which was also reflected in the discussion of the coding/analysis.

7. PLOS authors have the option to publish the peer review history of their article (what does this mean?). If published, this will include your full peer review and any attached files.

Reviewer #1: No

Reviewer #2: No

---

## [Editor Report · Acceptance letter]

21 Nov 2022

PONE-D-22-13001R1 

A Systematic Review of the 60 Year Literature: Effects of Outreach Programs in Supporting Historically Marginalized and First-Generation, Low-Income Students in Healthcare Education 

Dear Dr. Nardo:

I'm pleased to inform you that your manuscript has been deemed suitable for publication in PLOS ONE. Congratulations! Your manuscript is now with our production department. 

Kind regards, 

on behalf of

Dr. Nabeel Al-Yateem 

Academic Editor

PLOS ONE